# Bioprospecting for Rhizobacteria with the Ability to Enhance Drought Tolerance in *Lessertia frutescens*

**DOI:** 10.3390/ijms242417585

**Published:** 2023-12-18

**Authors:** Mokgadi M. Hlongwane, Felix D. Dakora, Mustapha Mohammed, Ntebogeng S. Mokgalaka-Fleischmann

**Affiliations:** 1Department of Chemistry, Tshwane University of Technology, Private Bag X680, Pretoria 0001, South Africa; hlongwanemm@tut.ac.za (M.M.H.); dakorafd@tut.ac.za (F.D.D.); 2Department of Crop Science, University for Development Studies, Tamale P.O. Box TL1882, Ghana; mmustapha@uds.edu.gh; 3Faculty of Science, Tshwane University of Technology, Private Bag X680, Pretoria 0001, South Africa

**Keywords:** abiotic stress, medicinal plant, metabolites, molecular networking, rhizobacteria, sutherlandiosides, sutherlandins

## Abstract

*Lessertia frutescens* is a multipurpose medicinal plant indigenous to South Africa that is used for the management of cancer, stomach ulcers, wounds, etc. The use and demand for the raw materials from this plant have been increasing steadily over the years, putting strain on the dwindling wild populations. Although cultivation may provide relief to the strained supply, the persistent drought climate poses a threat to the plant’s growth and productivity. This study explored three plant-growth-promoting rhizobacteria isolates, TUTLFNC33, TUTLFNC37 and TUTLFWC74, obtained from the root nodules of *Lessertia frutescens* as potential bioinoculants that can improve yield, biological activities and the production of secondary metabolites in the host plant. Isolate TUTLFNC37 was identified as the most promising isolate for inoculation of *Lessertia frutescens* under drought conditions as it induced drought tolerance through enhanced root proliferation, osmolyte proline accumulation and stomatal closure. Superior biomass yield, phenolics, triterpenes and antioxidant activity were evident in the extracts of *Lessertia frutescens* inoculated with TUTLFNC37 and under different levels of drought. Furthermore, the metabolomics of the plant extracts demonstrated the ability of the isolate to withstand drastic changes in the composition of unique metabolites, sutherlandiosides A–D and sutherlandins A–D. Molecular families which were never reported in the plant (peptides and glycerolipids) were detected and annotated in the molecular networks. Although drought had deleterious effects on *Lessertia frutescens*, isolate TUTLFNC37 alleviated the impact of the stress. Isolate TUTLFNC37 is therefore the most promising, environmentally friendly alternative to harmful chemicals such as nitrate-based fertilizers. The isolate should be studied to establish its field performance, cross infectivity with other medicinal plants and competition with inherent soil microbes.

## 1. Introduction

Drought remains one of the most detrimental abiotic stresses that impede plant growth and development [1]. Among other things, water scarcity in crops, and medicinal plants is known to negatively impact plant biomass [2,3,4,5], chlorophyll content [3,5,6] and relative water content [5,6]. Despite these negative effects, there is still little being done in pursuing studies that explore the effects of abiotic stress on medicinal plants and the mitigation thereof. This is a result of the misconception that abiotic stress in medicinal plants facilitates the accumulation of secondary metabolites, thereby improving their efficacy and growth [4].

Contrasting effects of drought on medicinal plants have been documented in several studies. For instance, Baher et al. [2] demonstrated how drought reduced plant yield in *Satureja hortensis* L., even though the essential oil yield was enhanced. Although conducive to essential oil production, drought stress compromised the plant biomass production. In another study, drought was found to significantly enhance the biomass yield of *Melissa officinalis* (lemon balm) and *Nepeta cataria* (lemon catmint), but not that of *Salvia officinalis* (sage) [7]. Moreover, polyphenols in lemon balm and sage were also increased due to drought, but no significant change was noted in lemon catmint [7]. The responses observed in medicinal plants under drought are not linear or easily predictable, indicating the complex impact of drought on different plant parameters and species. Given the differential response of plant species to drought, thorough species-specific studies are indispensable for optimal agronomic regimens that may be used for targeting specific outcomes. Furthermore, this evidence validates how reckless and careless blanketing statements may be.

Plant-growth-promoting rhizobacteria (PGPR) is a collective term for bacteria that improve plant growth through the production of plant hormones, facilitation of nutrient assimilation, suppression of ethylene synthesis and management of plant pathogens [8,9]. In their investigation, Gorgi et al. [5] demonstrated the ability of PGPR to restore yield, relative water content, photosynthesis parameters and proline in drought stricken lemon balm plants. Proline is a unique proteogenic amino acid with its α-amino acid group positioned as a secondary amine. Proline is mainly biosynthesised in higher plants from glutamate by 1-pyrroline-5-carboxylate synthetase and pyrroline-5-carboxylate reductase [10]. Production of proline is crucial for alleviating drought stress in plants. Proline is over-expressed by plants to increase osmolyte concentration and relieve the osmotic pressure in plants under drought. Although medicinal plants have the inherent ability to accumulate osmolyte solutes such as proline, choline and trehalose under drought, the production of these osmolytes is faster in PGPR [11]. Moreover, several authors have attributed the production of proline to the ability of a plant system to induce stress tolerance [5,6,12].

Canavanine together with sutherlandioside A–D and sutherlandin A–D are some of the important phytochemicals found in *L. frutescens* and are associated with the curative ability of the medicinal plant [13]. Although *L. frutescens* is relatively drought tolerant, Colling et al. [14] reported a slight reduction in the content of canavanine when plant seedlings were treated with 3% polyethylene glycol (PEG) which simulates drought. This observation is an indication that drought may result in the reduction of phytochemicals found in *L. frutescens*. However, it is still unclear whether the reduction will compromise the biological activities of *L. frutescens* extracts. In addition to the potential deleterious effects of drought on *L. frutescens* extracts is global warming which is being projected to worsen. This gloomy prospect is unfortunately accompanied by more people subscribing to the use of medicinal plants for safety and economic reasons [15]. Hence, there is constant increase in the demand for the raw materials of *L. frutescens*, which threatens to put more pressure on the dwindling supply of the wild populations. The propagation of *L. frutescens* is a potential solution to curb the growing demand for the plant and extinction due to overharvesting [16]. Given the current climate status, there will be a need to enhance the biomass yield and biological activities of medicinal plants. However, the conventional method to improve growth and development through the use of agrochemical fertilizers is marred with disadvantages [9,15,17]. Chemical fertilizers deplete soil fertility, nutrients, organic carbon and contaminate water bodies [15]. Moreover, agrochemical fertilizers are associated with exorbitant costs [18]. There is a great need for environmentally friendly interventions for cultivating *L. frutescens* with similar or better yields, antioxidant activities and phytochemical compositions under the prevailing climate conditions. This study seeks to investigate the effect of selected PGPR on the yield, biological activities and metabolites of *L. frutescens*.

## 2. Results

### 2.1. Plant Materials, Soil Preparation, Inoculation, and Planting

The effects of drought on the yield of *L. frutescens* inoculated with various treatments (TUTLFNC33 (I1), TUTLFNC37 (I2), TUTLFWC74 (I3), 5 mM potassium nitrate (KNO_3_), positive control (PosC) and negative control (NegC)) are illustrated in Figure 1. At the same time, Figure 2 shows the different growth rate of *L. frutescens* under moderate drought induced by different treatments. A two-way analysis of variance revealed a significant effect (*p* < 0.001) of the inoculation treatment, drought treatment as well as the inoculation × drought interaction on shoot biomass, root biomass, nodule mass and nodule number of the *L. frutescens* plants. For all quantitative parameters, the interaction effects are presented in this study. As shown in Figure 1a, I2 + LD induced the highest shoot biomass (428.67 ± 4.16 mg·plant^−1^), followed by I2 (417.0 ± 4.31 mg·plant^−1^) and then I2 + HD (398.1 ± 4.07 mg·plant^−1^). In general, inducing low drought significantly reduced the root biomass when compared to the unstressed counterparts. For instance, in plants inoculated with I2, the root biomass was 496.73 ± 2.31 mg·plant^−1^ when compared to the lower value of 154.20 ± 2.67 mg·plant^−1^ recorded in I2 + LD (Figure 1c). The further increase of drought to moderate (MD) and high (HD) levels was accompanied by an enhanced root biomass. This phenomenon was observed in all the treatments, with few exceptions (Figure 1c).

As expected, the plants treated with KNO_3_ solution and negative control did not elicit nodules. Interestingly, the positive control (PosC) plants which were inoculated with a commercial inoculant (*Bradyrhizobium* sp.) of groundnut and cowpea did not form nodules. The lowest shoot biomass was observed in the negative control (NegC) plants which were not supplied with nitrogen in the nutrient solution.

The highest root biomass was accumulated by plants inoculated with I2, while the lowest was recorded in plants treated with HD + PosC, LD + NegC, MD + Neg and PosC (Figure 1c). Again, the negative control and positive control exhibited the worst performance. The highest number of nodules was observed in plants inoculated with I3, while those inoculated with I2 recorded the lowest nodule number (Figure 1b). In addition to superior root proliferation, inoculation with I2 also resulted in the highest nodule biomass while the lowest nodule biomass was found in plants treated with HD + I1 (Figure 1d).

### 2.2. Morphological Characteristics

Micrographs for plants treated with I1, I3, KNO_3_, PosC and NegC were not included in this article, but are available upon request. However, the summary of the results encompasses all treatments. Pictographs of plants treated with I2 were shown as it outperformed other test isolates and has the potential to be developed further as a biological inoculum for *L. frutescens*.

The different treatments induced variable stomatal and morphological responses with increasing drought levels as visualised in the SEM micrographs in Figure 3. Across all treatments, plants without drought stress exhibited open stomata (Figure 3a). Plants treated with I2, positive control and negative control maintained open stomata when under low drought stress whereas the I1, I3 and KNO_3_ treatments induced partial stomatal closure under low stress. Moderate drought induced the closure or partial closure of stomata in all treatments, except for MD + I1, MD + I2, MD + KNO_3_ and MD + NegC. Under high drought conditions, plant leaves had their stomata closed, except for HD + I3. In addition to stomatal variances, the morphology of the leaves of plants under low and moderate drought presented a rough surface (Figure 3b,c), which could be attributed to low relative water content or dehydration. Meanwhile, the leaves without stress and at high drought were smooth.

The yields of *L. frutescens* plants treated with NegC and PosC were insufficient to be subjected to the assays and metabolomics analysis. Therefore, the following results do not contain data for the NegC and PosC. However, the KNO_3_ treated samples are employed as the positive control, since KNO_3_ is regarded a chemical fertilizer.

### 2.3. Osmolyte Proline Content Assay

The proline assay verified the ability of rhizobacteria to stimulate accumulation of osmolytes in plants under abiotic stress. All the tested rhizobacteria isolates were generally effective in inducing proline accumulation under drought when compared to the KNO_3_ treated plants. As depicted in Figure 4a, KNO_3_ treated plants fell short in the accumulation of proline, while plants treated with rhizobial isolates thrived. The highest proline was detected in plants treated with HD + I2, while lowest content was detected in plants treated with I3. Moreover, proline concentration increased with increasing level of drought, with some exceptions and the concentration was significantly lower in unstressed plants.

### 2.4. Total Flavonoid Compounds

The total flavonoid content was markedly influenced (*p* < 0.001) by the interaction effect of the inoculation treatment x drought. The flavonoid content of the plant tissues increased with the level of drought (Figure 4b), with exceptions. Interestingly, plants treated with I2 exhibited the lowest variation. The highest amount of flavonoids was detected in plants treated with MD + I1 at about 78.7 ± 0.71 mgQuE·g^−^ and the lowest in plants treated with I3 at about 27.7 ± 0.73 mgQuE·g^−1^. However, when compared among stress levels and inoculation treatments (i.e., I1 + LD, I2 + LD, I3 + LD, KNO_3_ + LD), plants treated with nitrate induced the highest amount of flavonoids, followed by plants treated with I1, I2, then lastly I3 (observed flavonoids content trend: KNO_3_ < I1 < I2 < I3).

### 2.5. Total Phenolic Compounds

In contrast to the flavonoids assay, the total phenolic compounds (TPC) exhibited relatively less variation within treatments (see Figure 4c). The highest amount of phenolics was reported in plants treated with HD + KNO_3_ at 31.7 ± 0.30 mgGaE·g^−1^ and lowest in MD + I3 at about 15.9 ± 0.23 mgGaE·g^−1^. The highest total phenolic content was found in the plant extracts treated with I2 under no stress, I2 under low drought, I1 under moderate drought and KNO_3_ under high drought. The lowest amounts of TPC for no stress, low drought, moderate drought and high drought were detected in the extracts of plants treated with KNO_3_, I3, I3 and I2, respectively. Ultimately, when considered holistically among stress levels, plants treated with I2 were superior in the synthesis of phenolic compounds. Subsequent treatments were I1, nitrate and I3, respectively (observed total phenolic content trend: I2 < I1 < KNO_3_ < I3).

### 2.6. Total Triterpenes

*Lessertia frutescens* treated with HD + I2 and HD + I3 exhibited the highest amount of triterpenes of about 2.50 ± 0.01 mg·g^−1^, while those treated with MD + KNO_3_ exhibited the lowest triterpenes equal to 1.50 ± 0.01 mg·g^−1^. The trend observed for the triterpenes content, which is depicted in Figure 4d is as follows: I2 < KNO_3_ < I3 < I1.

### 2.7. Antioxidant Activities

The antioxidant activity of *L. frutescens* extracts was determined using the DPPH and FRAP assays and the findings are depicted in Figure 5. In both protocols, the extracts of plants treated with HD + KNO_3_ exhibited superior antioxidant activity. Moreover, of the three rhizobacteria isolates tested, I2 was the most effective in inducing antioxidant activity when considered across all stress levels. In addition, there was consistency in the EC50 values for I2 across all drought levels, while nitrate and other test isolates induced a radical scavenging ability which increased with the drought intensity.

Radical scavenging activity results revealed that at 2 mg·g^−1^, ascorbic acid inhibited about 96% of the DPPH radical, at the same time, at the same concentration the HD + KNO_3_ samples inhibited 62% of DPPH. The highest activity for the three test isolates was observed in HD + I1 at an inhibition of 50%, HD + I2 at 51% and I3 at 47% inhibition.

*Lessertia frutescens* treated with KNO_3_ exhibited superior ferric reducing antioxidant power (FRAP) in comparison with all the rhizobacteria isolates (Figure 5b). The highest antioxidant potency was observed in HD, followed by LD, MD and then KNO_3_ only extracts with activity values of 0.46, 0.34, 0.31 and 0.30 mM ferrous equivalents, respectively. As illustrated in Figure 5b, FRAP content within the treatments generally increased with increasing stress levels. A contradicting trend was, however, noted in plants treated with I1. In this case, the FRAP content decreased as the stress level increased. These results demonstrate somewhat an inversely proportional relationship between proline content and total flavonoid content.

### 2.8. Metabolomic Composition of Drought Stressed Lessertia frutescens Treated with I2

The molecular network facilitated clustering of metabolites was detected in *L. frutescens*, which grouped according to their structural relatedness. Metabolites which had similar gas phase chemistries were clustered into molecular families provided their cosine score was ≥0.7 [19]. The computed molecular network resulted in 508 nodes (consensus spectra), where 204 nodes were clustered into 37 molecular family networks. The networks comprised a minimum of two nodes connected by an edge as per the GNPS spectral matching. Consensus spectra that were not clustered were shown as individual nodes with loops at the bottom of the molecular network (Figure 6). Regrettably, only 19 of the 204 clustered nodes were annotated, indicating the intricacy of the metabolites found in *L. frutescens* and inadequate reference spectra in the library database. The novel compounds for *L. frutescens*, sutherlandiosides A–D and sutherlandins A–D, belonging to the molecular classes of cycloartane triterpenoids and flavonoids are in cluster A and B of the network (Figure 6), respectively. Each of the metabolites was detected in all the tested *L. frutescens* extract samples. The corresponding molecular formulas, retention time and mass fragmentation are outlined in Table 1.

The compounds are derivatives of quercetin and kaempferol [20], hence the fragment ion of *m*/*z* ≈ 287 and *m*/*z* ≈ 303 Da (Table 1).

**Table 1 ijms-24-17585-t001:** Identification of sutherlandiosides A–D and sutherlandins A–D in *Lessertia frutescens extracts* inoculated with I2 and subjected to different drought levels. Fragmentation was verified against [21].

Molecular Formula	*m*/*z* Value	Retention Time (min)	Observed Fragmentation Ions	Compound Name	Samples
C_36_H_60_O_10_	652.4186	9.56	653.4315; 491.3750; 473.3657; 455.3556; 437.3448; 419.3339	Sutherlandioside A and B	I1, I2, I3, KNO_3_, LD + I1, LD + I2, LD + I3, LD + KNO_3_, MD + I1, MD + I2, MD + I3, MD + KNO_3_, HD + I1, HD + I2, HD + I3, HD + KNO_3_
C_36_H_58_O_10_	650.4030	8.85	651.4130; 633.4041; 489.3596; 471.3511	Sutherlandioside C	I1, I2, I3, KNO_3_, LD + I1, LD + I2, LD + I3, LD + KNO_3_, MD + I1, MD + I2, MD + I3, MD + KNO_3_, HD + I1, HD + I2, HD + I3, HD + KNO_3_
C_36_H_58_O_9_	634.4081	10.11	635.3539; 617.4042; 473.3637; 455.3539; 437.3432; 419.3326	Sutherlandioside D	I1, I2, I3, KNO_3_, LD + I1, LD + I2, LD + I3, LD + KNO_3_, MD + I1, MD + I2, MD + I3, MD + KNO_3_, HD + I1, HD + I2, HD + I3, HD + KNO_3_
C_32_H_36_O_20_	740.1800	6.91	763.1735; 741.1909; 609.1471; 302.9976	Sutherlandin A and B	I1, I2, I3, KNO_3_, LD + I1, LD + I2, LD + I3, LD + KNO_3_, MD + I1, MD + I2, MD + I3, MD + KNO_3_, HD + I1, HD + I2, HD + I3, HD + KNO_3_
C_32_H_36_O_19_	724.1851	7.20	747.1782; 725.1962; 593.1530; 287.0561	Sutherlandin C and D	I1, I2, I3, KNO_3_, LD + I1, LD + I2, LD + I3, LD + KNO_3_, MD + I1, MD + I2, MD + I3, MD + KNO_3_, HD + I1, HD + I2, HD + I3, HD + KNO_3_

Furthermore, quantitative descriptions of the novel metabolites are illustrated by the pie charts in Figure 6. Cluster A and B revealed little to no effect of the different drought levels (LD, MD, HD) on *L. frutescens* inoculated with I2. For instance, the *L. frutescens* metabolome was observed to broadly consist of similar proportions of the novel metabolites as indicated in Figure 6, cluster A and B. Some of the identified molecular families found in *L. frutescens* include glycerolipids, pseudoalkaloids and small peptides. There have not been studies that reported the presence of peptides and glycerolipids in *L. frutescens*. Thus, molecular networking serves as a rapid tool for the metabolomic screening of plants.

## 3. Discussion

Drought stress can reduce the yield and productivity of *L. frutescens*, a multipurpose medicinal plant with great potential for commercialization. Despite the damaging effects of drought on plants, the use of rhizobacteria to alleviate this abiotic stress is relatively under studied when compared to other stress factors such as salinity [22]. This study investigated the efficacy of three rhizobacterial strains (I1, I2 and I3) which were selected for their superior performance in in vitro assays for drought tolerance. Isolate I1 yielded the highest shoot biomass in the authentication experiment that tested ability of isolates to nodulate the homologous *L. frutescens* host, while I2 is the most drought resistant and I3 was the most salt tolerant. The three isolates together with a KNO_3_ solution, commercial inoculant (*Bradyrhizobium* sp.) and a negative control were used in *L. frutescens* subjected to no stress, LD, MD and HD levels of drought treatment. The findings revealed the varying effects of the test isolates on *L. frutescens* subjected to drought. In addition, an interesting phenomenon where KNO_3_ fed plants yielded lower shoot biomass than all the tested rhizobacteria isolates was observed. Similar observations have been reported in other studies [23]. Moreover, the commercial inoculant employed in the study failed to elicit nodulation in *L. frutescens* leading to low biomass production even though this inoculant has commonly been used for *L. frutescens*. For example, contrary to the findings of this study, Masenya et al. [24] and Makgato et al. [25] demonstrated the potential of the commercial inoculant in improving the yield, phytochemicals and antioxidant activities in *L. frutescens*. The insufficient yield observed in the negative and positive control plants resulted in the lack of data for the other tests conducted on the plants. However, KNO_3_ treated plants were used as a positive control to compare with the test isolates. The poor yield in the negative control samples can be attributed to the absence of nitrogen, an essential element for plant growth [26]. At the same time, the unexpected performance of the commercial inoculant in the current study can be attributed to possible incompatibility between the strain and the host *L. frutescens*. As the present study was carried out under sterile conditions, the studies by Masenya et al. [24] and Makgato et al. [25] may have benefited from the native soil microbial population rather than the inoculant. At the same time, Gerding et al. [27] hinted at the susceptibility of *Lessertia* species to ineffective nodulation by some *Rhizobium* bacterial species. These developments further underscore the urgent need for the formulation of an effective inoculum for the plant and the significance of this study. Studies attempting to alleviate the effects of abiotic stress on plants often take place in field conditions [28] or make use of matured plants [29]. In cases where they are conducted in sterile environments, similar results are evident where shoot biomass for the negative control is significantly lower than the treated plants [12]. In the case of the current study, the situation was further exacerbated by the miniature nature of *L. frutescens* seedlings. Isolate I2 elicited the highest and most consistent biomass yield in the host under the different abiotic stress levels, while other treatments resulted in relatively low biomass with significantly high variations when considered across the drought stress levels.

Plant response to abiotic stress is intricate, non-linear and species specific. Thus, the generalisation of responses or the use of one parameter in investigations is discouraged [30]. This study incorporated several parameters to establish which of the three test isolates would be a promising candidate for the inoculation of *L. frutescens* under drought. Critical characteristics that make a medicinal plant effective and feasible for commercialization include biomass yield, phytochemical composition and biological activities as these factors allow standardization for quality control [31]. A study by Chiappero et al. [29] which investigated the effect of plant-growth-promoting rhizobacteria on *Mentha piperita* grown under drought stress evaluated the total phenolics, biomass yield, antioxidant activities and proline content of the plants, among other parameters. Their findings revealed a detrimental effect of drought on plant growth. Moreover, the authors reported that drought stressed *Mentha piperita* inoculated with PGPR elicited significantly higher total phenolic content and antioxidant activities when compared to uninoculated counterparts. Similar observations were made in our study where the shoot biomass for stressed and uninoculated plants was significantly lower than those of their inoculated counterparts.

The mechanisms by which rhizobacteria promote growth in plants subjected to abiotic stress includes inducing root proliferation, proline accumulation and leaf transpiration reduction [22,30,32]. Ayuso-Calles et al. [33] corroborated this statement in their review by highlighting the inherent ability of PGPR to mitigate the osmotic pressure which is inevitable in drought stressed plants. According to Ayuso-Calles et al. [33], PGPR signals for the plants to accumulate osmolytes such as proline and improves soil porosity and aeration to maintain osmotic balance. Similar mechanisms for overcoming drought stress using PGPR were observed in this study. The leaves of *L. frutescens* characterized using SEM showed induced stomatal closure (Figure 3d) or transpiration reduction to maintain water content with increasing drought stress. Isolate I2, in particular, restored the smooth surface morphology of the leaves under the high drought condition by closing the stomata. A corresponding increase in plant biomass yield was recorded for plants inoculated with the isolate (Figure 1a).

The production of proline is crucial in stressed plants, as it functions to increase osmolyte concentration and relieve the osmotic pressure in plants. Although medicinal plants have an inherent ability to accumulate osmolyte solutes such as proline, choline and trehalose under drought, their production is faster in the presence of PGPR in the growth media [11,33,34]. The evaluation of proline content in stressed and inoculated plants in this study validated the ability of rhizobacteria to stimulate the accumulation of the osmolytes in plants under drought. The *L. frutescens* plants inoculated with the test rhizobacteria isolates accumulated higher levels of proline when compared to the plants treated with KNO_3_ solution (Figure 4a). The highest accumulation was noted in HD + I2 plants, which despite being subjected to severe drought, resisted a significant reduction in the shoot biomass yield. Several scholars associate proline accumulation in stressed plants to their ability to tolerate stress [12,35,36]. The more proline, the better the plant performance under stress conditions. Interestingly, Chiappero et al. [29] observed an opposite effect of drought on proline accumulation. The inoculation of drought stressed *Mentha piperita* with rhizobacteria in their study led to a decline in proline accumulation which the authors regarded as a positive outcome. The decline in proline content did not hinder the main goal of their study which was to induce drought tolerance by inoculation with rhizobacteria. The contradicting outcome emphasizes the complexity of plant processes and the importance of studying species rather than subscribing to general notions.

Ideally, the total phytochemicals and antioxidant activities of medicinal plants should yield consistent results for specimens sourced from different geographical locations, cultivated or wild and young or matured. However, it is evident that geographical origin, the nature in which a plant was established (wild or cultivated), age and other factors cause variations in the phytochemicals and antioxidant activities of plants [37]. Consequently, protocols which manage to produce consistent amounts of phytochemicals in medicinal plants are sought after due to their potential to expedite the processes of commercialization and standardization. Significant variation in critical phytochemicals may lead to overdose or underdose as medicinal plant users measure doses by considering the amount of raw material, negating the potential of phytochemical variation [38]. The total flavonoids in *L. frutescens* in this study demonstrated a trend where flavonoids increased as drought increased, with some exceptions (Figure 5a,b). Plant responses to abiotic stressors are complex and intricate [29], hence the differential response in total flavonoids observed in this study. In our study, the treatment of plants with rhizobacteria isolates resulted in enhanced flavonoid content (Figure 4b). The response is consistent with the findings of earlier studies which aimed to mitigate the negative effects of drought in plants using rhizobacteria [12,29]. Furthermore, the outcome of the antioxidant activity assay corroborated the fact that flavonoids are directly related to antioxidant activities [8,37]. In medicinal plants, the reduction of flavonoids or phenolics is not a positive trait as these phytochemicals are directly related to antioxidant activities and medicinal efficacy. Phenolic compounds are renowned for their ubiquitous antioxidant capability [39]. They are organic compounds with molecular structures consisting of one or more aromatic rings with one hydroxyl group. Overall, the TPC (Figure 4c) observed in *L. frutescens* within the treatments was fairly consistent when compared to the flavonoids (Figure 4b) which exhibited high variability within the treatments. Isolate I2 outperformed the other treatments with regard to the production of phenolics since it induced the highest TPC at two of the four drought levels (no stress and low drought) and second highest in one of the four drought levels (moderate drought).

The phytochemicals of *L. frutescens* include triterpenoid compounds which are defined as a collective term for all triterpenes [40,41]. Specifically, four triterpenoids glycosides (sutherlandiosides A, B, C and D) were isolated and identified in *L. frutescens* [42]. The triterpenoids glycosides were reported to have anticancer properties [43]. Moreover, sutherlandioside B was earmarked to be a biomarker for *L. frutescens* but was unfortunately discounted due to its geographical origin dependency [42]. The phytochemical is not always detected in *L. frutescens* specimens and this discrepancy is often observed among plant species originating from different localities. As far as the results for total triterpenes in this study are concerned, I2 was the most favourable for their production (Figure 4d). The triterpenes content peaked in moderate and high drought. Consistent with the flavonoids, the concentrations of triterpenes increased with increasing drought levels, with some exceptions. In both antioxidant activity assays, the plants treated with KNO_3_ were the best in synthesizing antioxidant compounds probably due to the two vital components of the treatment, potassium and nitrogen. Nitrogen is a critical macronutrient for plant growth and development in whole life process. Furthermore, nitrogen plays a vital role in the synthesis of amino acids, chlorophyll, phytohormones and phytochemicals. As a result, limited nitrogen in plants leads to plant senescence [44]. Potassium on the other hand is responsible for stimulating dozens of valuable enzymes involved in protein synthesis, sugar molecule transport, nitrogen and carbon metabolism, and photosynthesis. In addition, potassium is vital for cell growth, which is an important process for plant growth and development [45]. Despite the benefits of the KNO_3_ treatment in this study, the relationship between the total flavonoid content, total phenolic content and shoot biomass yield in plants treated with KNO_3_ conforms with the optimal defence hypothesis that states that a negative relationship exists between growth and defence in plants [46]. In essence when the compounds which are responsible for defending plants (flavonoids and phenolic compounds) increase due to stress levels, plant growth will be compromised. The results obtained in this study where plants were treated with I2 suggest that the isolate was effective in inducing stress tolerance. Hence the relatively low content of defence metabolites and high shoot biomass.

The screening of the sutherlandiosides A–D and sutherlandins A–D in the *L. frutescens* extracts inoculated with I2 under different drought conditions indicated that the isolate induced drought tolerance in the plant by withstanding the significant reduction or complete disappearance of a metabolite (Figure 6A,B). Similar observations were made in the extracts inoculated with I1, I3 and KNO_3_. The quantification of these metabolites remains a challenge as the compounds are not commercially available [20,47]. In the absence of commercial standards, the molecular network has become an indispensable annotation tool to provide semi-quantitative data to evaluate the impact of abiotic stress and treatment on the metabolite profiles of plants [48]. Therefore, the quantification information provided by the pie charts in (Figure 6) is adequate to make conclusive remarks. The metabolites of interest (sutherlandiosides A–D and sutherlandins A–D) detected in all the *Lessertia frutescens* extracts under various drought conditions (depicted by the four node colours) were somewhat consistent, except for content. Moderate drought (yellow pie) negatively affected the amount of sutherlandins A and B in *Lessertia frutescens.* This observation is consistent with the results obtained in the same sample (MD + I2) for shoot biomass (Figure 1a), leaf morphology (Figure 3), proline content (Figure 4a), total flavonoid compounds (Figure 4b) and total phenolic compounds (Figure 4c). In all the mentioned analyses, the plants under moderate drought yielded inferior results when compared to the other drought levels (no stress, LD and HD), with few exceptions.

## 4. Materials and Methods

### 4.1. Source of Bacterial Isolates and Commercial Inoculant

The three plant-growth-promoting rhizobacteria used in this study were earlier isolated from the root nodules of *L. frutescens* collected in the Northern Cape (NC) and Western Cape (WC) provinces of South Africa. The GPS coordinates for the locations where rhizosphere soils were collected are −31.966630, 20.312850 for TUTLFNC33 (I1) and TUTLFNC37 (I2); and −33.990065, 18.427481 for TUTLFWC74 (I3). The test isolates were earlier authenticated for their ability to induce root nodules on the homologous *L. frutescens* host followed by their selection based on performance in drought, pH and salt stress tolerance assays. Moreover, data based on the 16S rRNA gene sequences show that the three test isolates share some level of similarities with different rhizobial species in the genus *Mesorhizobium*. The 16S rRNA gene sequences have been deposited in GenBank under the accession numbers OR936566 for isolate TUTLFNC33 (I1), OR936573 for TUTLFNC37 (I2) and OR936558 for isolate TUTLFWC74 (I3). A commercial inoculant of groundnut (*Arachis hypogea*) and cowpea (*Vigna unguiculata*) containing *Bradyrhizobium* sp. was used as a positive control (PosC); this was a kind gift from Gert Rossouw, UPL Ltd., Kempton Park, South Africa.

### 4.2. Plant Materials, Soil Preparation, Inoculation, and Planting

The experiment involved four levels of drought, namely no stress (NS), low drought (LD), moderate drought (MD) and high drought (HD) and six levels of inoculation treatments namely (isolate I1, I2, I3, PosC, 5 mM KNO_3_ and a negative control supplied with all nutrients except nitrogen (NegC). Thus, there were a total of 24 treatment combinations which were arranged in a randomized complete block design with three replications per each treatment combination. The experiment was conducted in plastic pots under a naturally lit glasshouse for a period of 90 days.

Before planting, *L. frutescens* seeds were manually scarified, surface-sterilized and planted in sterile sand. Seven day old seedlings were inoculated with respective treatment. For the plant-growth-promoting isolates, 1 mL broth suspension of pure colony bacterial culture grown to exponential phase (10^6^–10^7^ cells/mL) was used. Meanwhile, a 5 mM KNO_3_ solution and uninoculated plants were included as positive and negative control, respectively. A sterile suspension of PosC was used to inoculate the plants designated as positive control. The treated seedlings were fed with sterile N-free nutrient solution [49] and sterile distilled water in alternation. After 60 days, when the plants were well developed, different stress levels were induced by watering the seedlings with 100 mL deionized water for no stress, 4% PEG 6000 for low drought (LD), 8% PEG 6000 for medium drought (MD) and 16% PEG 6000 for high drought (HD). The experiment proceeded with the same watering regime for 90 days after which the plants were harvested. Roots and shoots biomass were estimated by freeze drying the roots and shoots for 48 h and weighing the dry biomass. Fresh nodules were counted and weighed immediately after harvest.

### 4.3. Morphological Characteristics

A day before harvesting, five *Lessertia frutescens* leaves were carefully plucked and placed in a 24-well cell culture plate. The leaves were fixated overnight, then washed three times with a buffer. Graded ethanol ranging from 30–100% was used to dehydrate the leaves. Once the ethanol was removed, 50:50 mixture of hexamethyldisilazane (HMDS) and absolute ethanol was added and retained for an hour followed by soaking in concentrated HMDS. The samples were then covered and left under a fume hood for an hour. The leaves were mounted onto an aluminium stub covered with carbon tape and then coated with carbon. A Zeiss field emission gun scanning electron microscope (SEM) (Zeiss Gemini 55 Ultra Plus FEGSEM, Oberkoche, Germany) was used to study morphological changes in leaf tissues of *L. frutescens*.

### 4.4. Osmolyte Proline Content

Proline was determined in fresh leaves according to a method by Bates et al. [50]. About 0.5 g of wet leaf sample was homogenized in 3% sulphosalycylic acid and filtered using a filter paper. The filtrate was mixed with ninhydrin and glacial acetic acid. The mixture was heated for 60 min at 100 °C and upon completion, the reaction was quenched with ice and extracted with toluene. The absorbance of toluene extract was measured at 520 nm using a microplate reader (SpectraMax 190, Sunnyvale, CA, USA). A standard calibration curve was constructed using proline and used to estimate the analyte concentration in μg·g^−1^ FW. The analysis was performed in triplicate.

### 4.5. Total Flavonoid Compounds

Total flavonoids were determined by the spectrophotometric method following a method by Sulaiman and Balachandran [51] with minor modifications. Exactly 100 µL of double deionised (dd) water (blank) or quercetin standard (calibration from 0 to 1 mg·mL^−1^) or plant crude extract (5 mg per 1 mL solvent) was mixed with 300 µL of dd water in 2 mL Eppendorf tubes. Subsequently, 30 µL of 5% NaNO_2_, 30 µL of 10% AlCl_3_ and 200 µL of 1 M NaOH were added in intervals of five minutes. The tubes were vortexed to mix the contents. All the preparations were performed in triplicate. The vortexed samples were pipetted into a 96 well ELISA^®^ plate for reading the absorbance at 510 nm using a microplate reader (SpectraMax 190, Sunnyvale, CA, USA). The concentrations were reported as quercetin equivalent (QE) milligrams per gram of crude extract.

### 4.6. Total Phenolic Compounds

Total phenolic compounds (TPC) in the crude extracts of *L. frutescens* were determined using the *Folin–Ciocalteu* reagent method adapted from Ahmed et al. [52] with minor alterations. In a 2 mL Eppendorf tube, exactly 20 µL sample (5 mg crude extract per 1 mL solvent) was mixed with 1580 µL double deionised (dd) water, 300 µL freshly prepared Na_2_CO_3_ solution and 100 µL *Folin–Ciocalteu* reagent. The resulting mixture was wrapped with a foil and incubated at room temperature for 2 h. Blank sample was prepared with 20 µL dd water. The samples were accurately transferred to the 96 well ELISA^®^ plate for spectrometric absorbance determination at 765 nm using a microplate reader (SpectraMax 190, Sunnyvale, CA, USA). The samples and standards were prepared in triplicate. A series of 8 calibration standards ranging from 0 to 1 mg·mL^−1^ gallic acid were used to construct a standard calibration curve. Concentration observed from the calibration curve was reported as gallic acid equivalent (GAE) milligrams per gram of crude extract.

### 4.7. Total Triterpenes

Total triterpenes content was estimated through the colorimetric method according to [53]. A 75 µL methanol (blank) or 75 µL *L. frutescens* crude extract (5 mg/mL methanol) or relevant aliquot of β-sitosterol standard to prepare six calibration standards ranging from 0 to 5 mg·mL^−1^ were transferred into test tubes. Blank, samples and standards were all prepared in triplicate. The tubes were kept in a water bath at 85 °C until dryness, followed by addition of 250 µL vanillin solution (50 mg·mL^−1^). Then, 500 µL of sulfuric acid was added to the tubes and heated at 60 °C in a water bath for 30 min. The mixture was cooled in an ice bath prior to addition of 2500 µL acetic acid. The solutions were kept at room temperature for 20 min. Exactly 200 µL of each sample was pipetted in a well ELISA^®^ plate and absorbance was read at 548 nm using a microplate reader (SpectraMax 190, Sunnyvale, CA, USA).

### 4.8. Antioxidant Activities

Two methods we employed for the assessment of antioxidant activity in the *L. frutescens* extract. Radical scavenging activity which involves 2,2-diphenyl-1-picrylhydrazyl radical (DPPH•) and ferric reducing antioxidant potency which hinges on the reduction of ferric-tripyridyltriazine to its ferrous complex.

#### 4.8.1. Radical Scavenging Activity

The radical scavenging activity (RSA) of the different *L. frutescens* samples was estimated by making minor modifications to the DPPH method which Baliyan et al. [54] employed. The reagents used were all freshly prepared and kept in the dark throughout the experiment. A 1 mg·mL^−1^ methanolic solution of DPPH was used for the assay, while ascorbic acid was employed as a positive control. Ascorbic acid standards ranging between 0.01 and 0.2 mg·mL^−1^ and *L. frutescens* extract samples ranging between 0.025 and 1 mg·mL^−1^ were prepared. In a 96 well ELISA^®^ plate, 200 μL deionized water, 20 μL sample/positive control/blank and 90 μL DPPH solution was dispensed into each well. The contents of the plate were vortex, covered in aluminium foil and then incubated at room temperature for 30 min. Subsequently, absorbance reading was made at 517 nm using a microplate reader (SpectraMax 190, Sunnyvale, CA, USA). The assay was performed in triplicate.

The RSA of the extracts was calculated using the formula below:%RSA = [(CA − SA) ÷ CA] × 100
where CA is absorbance of the DPPH working solution, and SA is the absorbance of samples or ascorbic acid. The antioxidant activity of the sample extracts was reported as EC50 which is the amount of extract in question required to reduce the initial content of DPPH by 50%.

#### 4.8.2. Ferric Reducing Antioxidant Potency

Ferric reducing antioxidant potency (FRAP) assay kit (Sigma Aldrich, St Louis, MO, USA) and protocol was used to determine the ferric reducing capacity of *L. frutescens* extracts. A 5 mg·mL^−1^ *L. frutescens* extract was prepared for each sample, 2 mM ferrous stock was used as a standard and FRAP positive standard as a positive control. Relevant amount of stock was dispensed in respective wells of 96 well ELISA^®^ plate to prepare 0, 0.02, 0.04, 0.06, 0.08 and 0.10 mM ferrous standards. A FRAP buffer was used to make up the volume to 10 µL. Exactly 4 µL positive control plus 6 µL buffer was dispensed into a well, while 10 µL sample extracts were used instead. A reaction mixture (190 µL) was added to all the wells followed by incubation in a dark room for 60 min. The analysis was performed in triplicate. The absorbance reading was made using a microplate reader at 594 nm. The FRAP was reported as mM ferrous equivalents per mg crude extract.

### 4.9. Untargeted Metabolomics of Lessertia frutescens under Drought from Ultra-High Performance Liquid Chromatography Quadruple Time of Flight Mass Spectrometry

Samples for metabolomics analysis were prepared in single replicate by dissolving 5 mg crude extract in 1 mL methanol and then sterile filtered with a 0.22 µL syringe membrane into 1.5 mL sample vials fitted with 500 µL inserts. The samples were stored in 4 °C until analysis. Exactly 3 µL of *Lessertia frutescens* methanolic extracts for each sample was injected into the ultra-high performance liquid chromatography quadruple time of flight mass spectrometer (UHPLC-qTOF-MS) (LCMS-9030 qTOF, Shimadzu Corporation, Kyoto, Japan) for data acquisition. The analysis employed a protocol outlined by Ramabulana et al. [19] where chromatographic separation was performed in a Shim-pack Velox C18 column (100 mm × 2.1 mm with particle size of 2.7 µm) (Shimadzu Corporation, Kyoto, Japan) maintained at oven temperature of 55 °C. A binary mobile phase gradient system comprising solvent A (0.1% formic acid in Milli-Q water) and solvent B (methanol with 0.1% formic acid) was used. A gradient run at a flowrate of 0.4 mL.min^−1^ was set up to begin with 10% solvent B for 3 min, followed by 10–60% B over 3–40 min, 60% B from 40 to 43 min, and 90% B from 43 to 45 min (maintained for 3 min), returning to initial conditions from 48 to 50 min, followed by a 3 min column re-equilibration time.

The chromatographic products were further analysed utilizing the quadrupole time of flight high-definition mass spectrometer operating at negative electrospray ionisation mode with detector voltage of 1.8 kV. Additionally, interface voltage of 4.0 kV, interface temperature of 300 °C, nebulization and dry gas flow of 3 L·min^−1^, heat block temperature of 400 °C, desolvation line (DL) temperature of 280 °C and the flight tube temperature of 42 °C were used. Sodium iodide was used as a calibration solution to maintain high mass accuracy. Data dependent MS1 and MS2 were generated simultaneously for all ions of mass units between 100–1000 Dalton (Da) surpassing an intensity threshold of 5000. Fragmentation was executed using argon as a collision gas at a collision energy of 30 eV with a spread of 5 eV.

#### 4.9.1. Molecular Networking and Metabolite Annotation

Raw output from the UHPLC-qTOF-MS was converted to an open-source format (mzML) prior to being uploaded to an online workflow on the Global Natural Products Social (GNPS) website. Molecular networks were then constructed using the online workflow (https://ccms-ucsd.github.io/GNPSDocumentation/) (accessed on 4 October 2023). The uploaded data was filtered by removing all MS/MS fragment ions within ±17 Da of the precursor m/z and choosing only the top 6 fragment ions in the ±50 Da window throughout the spectrum. The data was clustered using an MS-CLUSTER algorithm. The precursor ion mass tolerance and the MS/MS fragment ion were set to 0.05 Da. Edges which are similarity interconnections between metabolites were formed only if the cosine score of 0.7 was exceeded with more than 6 matched peaks. The network TopK was set to 10 of which only nodes in each other’s respective top 10 similar nodes were kept in the network. The maximum number of nodes that can be connected into a single molecular family was set at 100, while the lowest scoring edges were removed. Spectra in the network were searched across spectral databases such as MassBank, ReSpect and NIST where the library spectra were filtered in the same manner as the sample spectra. The generated molecular networks were viewed using Cytoscape software version 3.8.1. All matched and some unmatched nodes were verified or putatively annotated using empirical formulae generated from accurate mass and fragmentation patterns obtained from MS/MS experiments. SIRIUS software version 5.8.3 was used for the analysis of the metabolites that integrate the CSI:FingerID with COSMIC, ZODIAC, and CANOPUS web service. Compounds were identified for sample processing by combining precise mass measurements with an internal retention time [55]. The compounds were also verified against various natural product dereplication databases such as KNApSAck, ChemSpider, PubChem, and Dictionary of Natural Products. These were also compared to the available literature. Metabolite annotation was carried out at level 2 of the Metabolomics Standards Initiative (MSI) [56].

#### 4.9.2. Metabolomics Identification of Sutherlandiosides A–D and Sutherlandins A–D in Drought Stricken *Lessertia frutescens*

Putative metabolite identification of the novel metabolites of *Lessertia frutescens* (sutherlandiosides A–D and sutherlandins A–D) was undertaken using bioinformatics. The mzML format output was exported to SIRIUS followed by two steps to annotate the unidentified metabolites. The compounds were searched using the SIRIUS bioinformatic tool by computing the data, verifying and comparing the proposed molecular formula and mass fragmentation against the available literature on the metabolites [48]. The retention time for the verified metabolite candidates were used to search presence of the metabolites in samples inoculated with I2 and subjected to different drought levels to establish whether there was a major change on the metabolomic composition of *L. frutescens* extracts triggered by the different conditions.

### 4.10. Statistical Analysis

The results for total phenolics, total flavonoids, proline, total triterpenes and FRAP are presented as mean ± standard error (SE). The results were subjected to a two-way analysis of variance (ANOVA) to determine statistical differences resulting from inoculation x drought treatment interactions after normality testing. Differences that were found at *p* ≤ 0.05 between means were considered statistically significant. Duncan’s multiple range test was used to separate means which were significantly different. The EC50 values for DPPH reduction were calculated on GraphPad Prism 7.04, within the dilution series that was prepared. Plant extract concentrations at which 50% cell growth is inhibited (EC50) were determined by the log sigmoid dose–response curve plotted on GraphPad Prism 4.

## 5. Conclusions

Drought stress may have negative and positive effects on plants. There is, therefore, a need for researchers to strike an optimal balance to ensure sustainability, productivity and the reliability of literature which suggests abiotic stress can be used to increase the phytochemical content and antioxidant activity of medicinal plants. *Lessertia frutescens* inoculated with rhizobacteria isolates tolerated drought stress through several mechanisms including root growth, osmolyte proline production and transpiration rate reduction or stomatal closure. These responses further resulted in enhanced biomass yields, improved biological activities and phytochemicals.

Rhizobacterial isolate I2 induced the highest yield, proline, phenolics and triterpenes in plants. Furthermore, the same isolate (I2) outperformed test rhizobacteria I1 and I3 on the antioxidant activity assays. In addition, the molecular network for the extracts inoculated with I2 revealed consistent levels of the unique metabolites of *L. frutescens* (sutherlandiosides A–D and sutherlandins A–D) in extracts under different drought levels. This observation highlights the surpassing ability of the isolate to induce drought tolerance in the medicinal plant. Therefore, isolate I2 is the most promising environmentally friendly alternative to harmful chemicals such as nitrate-based fertilizers. Thus, isolate I2 should be explored further to establish field performance, cross infectivity with other medicinal plants and competition with inherent soil microbes.

## Figures and Tables

**Figure 1 ijms-24-17585-f001:**
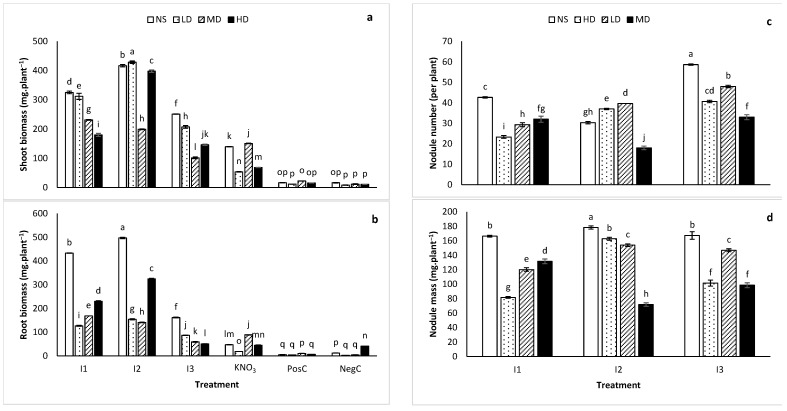
(**a**) Shoot biomass, (**b**) root biomass, (**c**) nodule number (NN) and (**d**) nodule mass (NM) of *Lessertia frutescens* treated with I1, I2, I3, KNO_3_, PosC and NegC while subjected to no stress (NS), moderate drought (MD) and high drought (HD). Bars with different letters within a column are significantly different at *p* < 0.001, *n* = 3. Error bars represent standard errors.

**Figure 2 ijms-24-17585-f002:**
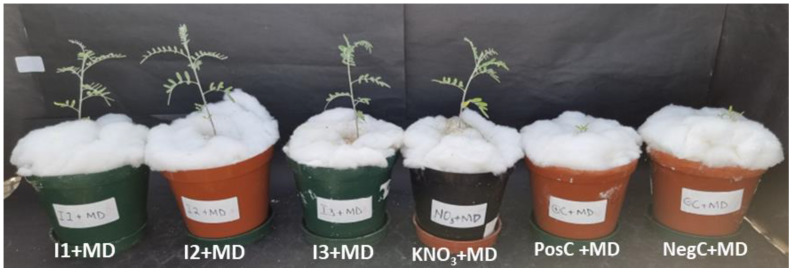
The growth rate of *Lessertia frutescens* plants subjected to treatments I1, I2, I3, KNO_3_, PosC and NegC under moderate drought at 90 days after planting.

**Figure 3 ijms-24-17585-f003:**
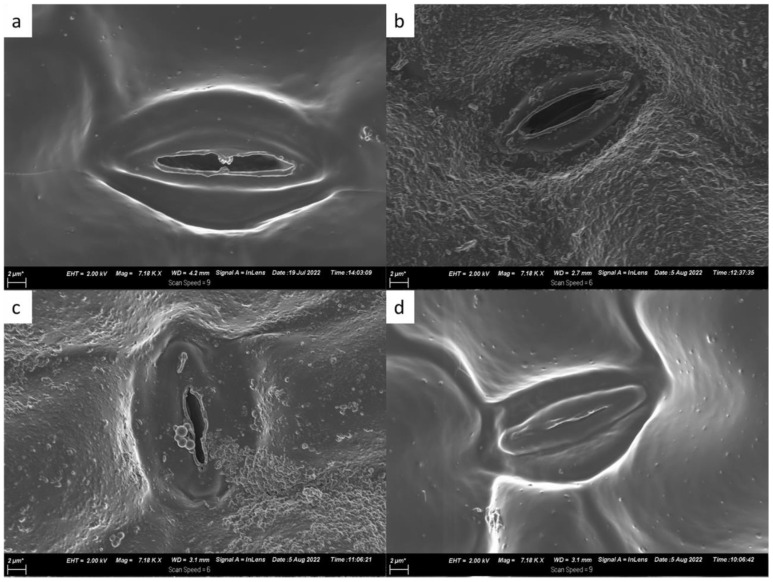
Scanning electron microscopy images for the stomatal aperture of plants inoculated with I2 under (**a**) no stress, (**b**) low drought, (**c**) moderate drought and (**d**) high drought.

**Figure 4 ijms-24-17585-f004:**
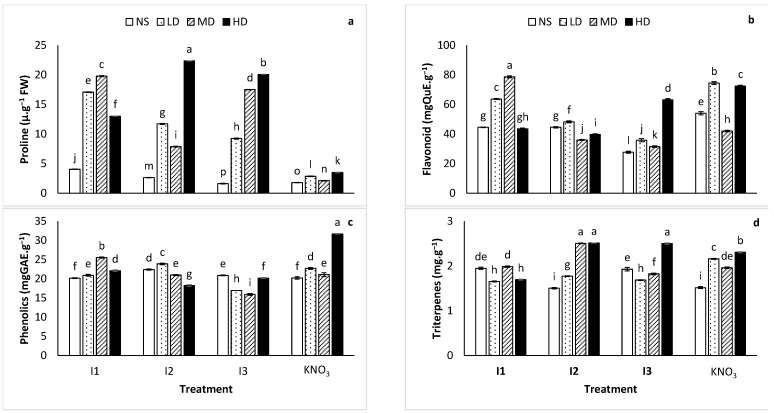
Variation in (**a**) proline, (**b**) total flavonoids content, (**c**) total phenolics content and (**d**) total triterpenes content in *Lessertia frutescens* treated with rhizobacteria I1, I2 and I3; and KNO_3_ under different drought levels (NS, non-stressed; LD, low drought; MD, moderate drought and HD, high drought). Bars with dissimilar letters are significantly different at *p* < 0.001, *n* = 3. Error bars represent standard errors.

**Figure 5 ijms-24-17585-f005:**
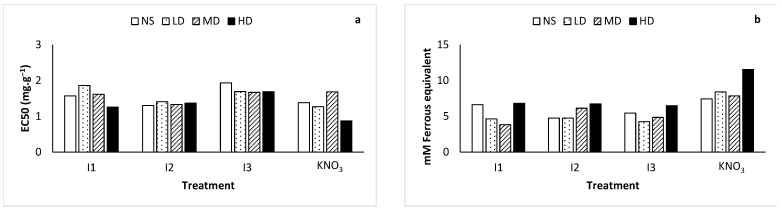
Antioxidant activity of *Lessertia frutescens* estimated through radical scavenging activity presented as (**a**) EC50 values and (**b**) ferric reducing antioxidant power. The plants exposed to no stress (NS), low drought (LD), moderate drought (MD) and high drought (HD) were treated with rhizobacteria I1, I2, I3 and KNO_3_ solution.

**Figure 6 ijms-24-17585-f006:**
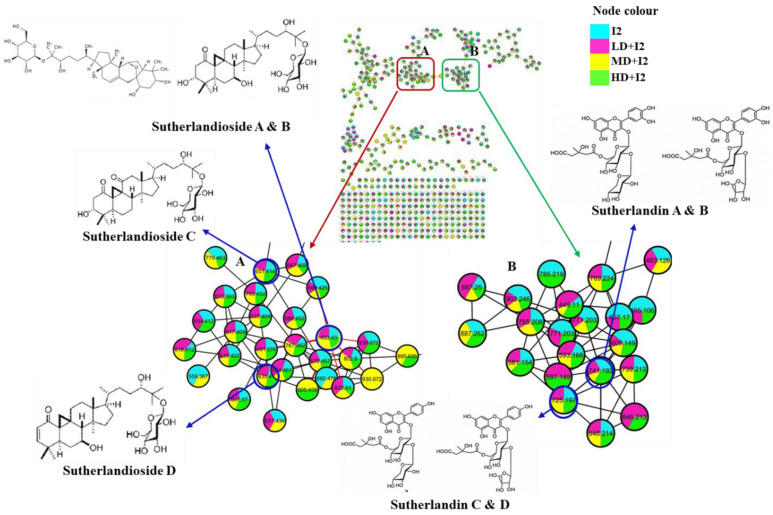
Molecular network of *Lessertia frutescens* extracts inoculated with I2 subjected to different drought levels, no drought (I2), low drought (LD + I2), moderate drought (MD + I2) and high drought (HD + I2). The extracts were analysed by ultra-high performance liquid chromatography quadruple time of flight mass spectrometry with two molecular families, cycloartane triterpenoids (A) and flavonoids (B). Node colours represent the level of drought which the extracts were subjected to.

## Data Availability

Data are contained within the article.

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
