# Peer review of "Bioprospecting for Rhizobacteria with the Ability to Enhance Drought Tolerance in Lessertia frutescens"

_ijms, 2023, doi:10.3390/ijms242417585_

Round 1

Reviewer 1 Report

Comments and Suggestions for Authors

Dear Authors

The present manuscript entitled “Bioprospecting for a rhizobacteria with the ability to enhance  drought tolerance in Lessertia frutescens” explored three plant growth promoting rhizobacteria isolates (isolate 1, 2 and 3 obtained from rhizosphere of Lessertia frutescens) for the superior environmentally friendly bioinoculant that can improve yield, biological activities, and production of secondary metabolites in Lessertia frutescens. Results suggested Isolate 2 is the most promising, environmentally friendly alternative to the harmful chemicals such as nitrate-based fertilizers. The study is very well planned and presented nicely, although there are some minor queries and suggestions. Line 167- Please explain why the yields of L. frutescens plants treated with NegC and PosC were insufficient? If there was no growth observed? There would be nice to investigate key genes and their expression during the interaction. The conclusion section may be short and more concise to express the key message of the present study.

Thank you

Reviewer 2 Report

Comments and Suggestions for Authors

The topic of growth-promoting organisms is currently very popular among scientists and the reviewed work undoubtedly fits into this global trend. Therefore, the authors took up an important issue and collected a lot of material, and as the work shows, they still have data that has not been presented (and is available for possible viewing?). Undoubtedly, the strong point of the work is the large number of results of chemical analyses, and in principle I can say that the part of the work that determines the role of stress in shaping the content of important compounds in the tested plant is very positive. However, I have some doubts regarding the de facto main factors of experiments  (isolates) I1, I2 and I3, and in particular their identification. What would a possible repetition of the research involve? In fact, my positive opinion about the paper as whole depend on the answer to this question.
And a few minor comments:
- among keywords there are no rhizobacteria
- in description of results unnecessary some literature inclusions
- conclusions  - literature data are not need
-microscopic photos are interesting, but in order to become reliable evidence in scientific work, they must be accompanied by a broader description.
